# LOG PROBABILITY TRACKING OF LLM APIS

**Timothée Chauvin** [1]    **Erwan Le Merrer** [1]    **François Taïani** [1]    **Gilles Tredan** [2]

[1]Université de Rennes, Inria, CNRS/IRISA (Rennes, France)
[2]LAAS, CNRS (Toulouse, France)

## ABSTRACT

When using an LLM through an API provider, users expect the served model to remain consistent over time, a property crucial for the reliability of downstream applications and the reproducibility of research. Existing audit methods are too costly to apply at regular time intervals to the wide range of available LLM APIs. This means that model updates are left largely unmonitored in practice. In this work, we show that while LLM log probabilities (logprobs) are usually non-deterministic, they can still be used as the basis for cost-effective continuous monitoring of LLM APIs. We apply a simple statistical test based on the average value of each token logprob, requesting only a single token of output. This is enough to detect changes as small as one step of fine-tuning, making this approach more sensitive than existing methods while being 1,000x cheaper. We introduce the TinyChange benchmark as a way to measure the sensitivity of audit methods in the context of small, realistic model changes.

## 1 INTRODUCTION

LLM API providers typically offer version-pinned endpoints, signaling to users that a given endpoint will serve a consistent model. Users of APIs tend to rely on this consistency: *developers* want to avoid unexpected regressions in their applications; *researchers* seek reproducibility in their experiments; *regulators* perform initial compliance assessments, and assume that the API will keep serving the same model afterward (Yan & Zhang, 2022). Yet users have no practical way to verify this consistency. Once a niche concern, this gap is drawing more attention as LLMs get deployed in mission-critical applications, notably in software engineering.

LLMs are, however, highly complex software artifacts under constant development. LLM API providers may change their inference software and hardware infrastructure for performance reasons, or modify the model itself in response to new jailbreaks or to update its behavior. Some providers might also quietly deploy quantized versions to save costs, switch to more light-weight models during peak traffic, or even serve different variants based on query characteristics. More concerning from a security perspective, providers might suffer the malicious injection of unobstrusive model backdoors, infringing the privacy and security of their users for an attacker's benefit. Such explicit attacks could happen from insiders, a bad update, or a compromised supply chain. This concern isn't just theoretical: the version of Grok deployed on X had three such incidents in 2025 where a modified system prompt was deployed to users of the X platform. Two incidents were blamed on rogue employees (Babuschkin; xAI, b) and one on a bad update (xAI, a).

Unfortunately, existing change-detection methods tend to be prohibitively expensive. State of the art approaches typically rely on the processing of tokens returned by the monitored LLM (Chen et al., 2023; Gao et al., 2025; Cai et al., 2025) and require extensive benchmarking or statistical analysis across many queries to reach a robust conclusion. As a result, LLM APIs are left largely unmonitored by third parties for changes in practice.

In this paper, we propose to address these shortcomings by exploiting the log probabilities of the returned tokens rather than the tokens themselves, a method we call *logprob tracking* (**LT**). During LLM inference, each token is sampled from an underlying vector of log probabilities (logprobs) over

---

Code available at: `https://github.com/timothee-chauvin/track-llm-apis`

the entire vocabulary space of the model. Some LLM APIs can return a subset of these logprobs for each token of the response. While not systematic, a nontrivial fraction of LLM APIs support logprobs. For instance, we found that 23% of reachable endpoints on OpenRouter—a widely used gateway to 60+ LLM API providers and 500+ models—return logprobs when requested.

Intuitively, logprobs constitute a far richer information source than tokens, yielding a handful of continuous values along with a token. Unfortunately, exploiting these logprobs is not trivial. Logprobs are not deterministic in practice (Cai et al., 2025), making them more challenging to use for model change auditing than simply checking logprob vectors for equality across time.

In this work, we show that log probabilities reveal enough information to detect even minor changes in LLM APIs, such as a single step of fine-tuning. This makes our approach significantly more sensitive than existing methods. Specifically, we use simple statistical methods to overcome logprob non-determinism, effectively detecting changes across a wide range of representative model modifications. We demonstrate through extensive in-vitro experiments that sending a single token of input and requesting a single token of output is enough to consistently detect changes that remain undetected by current monitoring methods. Our input token is chosen arbitrarily, and unrelated to the type of change applied. We report that our approach is up to three orders of magnitude cheaper than existing alternatives.

In this paper, our main contributions are the following:

- We introduce the *logprob tracking* (**LT**) method, and show that a 1-token prompt and the logprobs of a 1-token response are sufficient to exceed the detection performance and sensitivity of alternative methods, at a fraction of the cost;

- We introduce the TinyChange benchmark, designed to evaluate detection methods on small model changes;

- We extensively evaluate **LT** using the TinyChange benchmark against a wide range of representative model changes (finetuning, random noise, pruning), and compare it to two state-of-the art approaches, demonstrating how LT is able to detect changes as small as one step of fine-tuning at a fraction of the monitoring cost of other methods.

The remainder of the paper is organized as follows: in section 2 (Method), we describe the problem setting and the intuition for using logprobs for change detection. We introduce a statistical test to work around the non-determinism of logprobs. In section 3 (Experiments), we introduce the TinyChange benchmark, designed to evaluate change detection methods on small, realistic changes. We use it to show that prompts as small as a single token are almost as good as longer prompts for detection. We show that LT significantly exceeds the detection accuracy of state-of-the-art baselines, while being orders of magnitude cheaper.

## 2 METHOD

### 2.1 RATIONALE

**Problem setting**   In practice, our goal is to identify systematic modifications that affect the API's output distribution, whether from model changes (fine-tuning, further post-training, quantization, pruning, system prompt modification, introduction of a backdoor...), or software/hardware changes (CUDA versions, inference kernels, set of GPUs used for inference...). We do not attempt to distinguish between the sources of change, as any systematic difference could impact reproducibility of research and application reliability, regardless of its origin.

We follow the definition of the model equality testing problem in Gao et al. (2025): we consider a pair of LLM APIs (which can be the same API at different times).

For a given input distribution of prompts, the auditor has sample access to the distributions over completions of both APIs, $P$ and $Q$. The audit aims to test if $P = Q$: two APIs are considered equivalent if and only if their response distribution is the same for all prompts.

**Logprobs**   Almost all modern LLMs are based on auto-regressive transformers. When processing an input, the output of the transformer is a vector of logits, with one logit for each token in the

tokenizer's vocabulary. These logits are then converted to a vector of probabilities through a soft-max operation, and the generated token is sampled from this probability vector. Log probabilities (logprobs) are simply the logarithm of each of these probabilities.

Some LLM APIs support returning the top-$k$ highest logprobs in addition to the sampled tokens. For instance, 23% of reachable endpoints on OpenRouter support returning between 5 and 20 logprobs. This includes all available models from xAI, multiple models from OpenAI including the GPT-4.1 series, and multiple open-weight models such as the Qwen, Gemma, Llama, Deepseek, or Mistral families of models. More details in Appendix A.1.

These logprobs encode a part of the probability distribution, and as such are denser in information than the sampled tokens. For instance, they can be used to recover the prompt tokens given an output (Morris et al., 2023), and have been used to leak proprietary information about an LLM, including its hidden dimension and embedding projection layer (Finlayson et al., 2024; Carlini et al., 2024).

This motivates us to study the feasibility of detecting changes in LLMs via the top-$k$ logprobs of a single output token, on a fixed short prompt. Intuitively, detecting many types of change using only a single prompt and a single output token seems possible, as LLM modifications, such as finetuning or quantization, typically affect all the weights of a model. In particular, finetuning in one domain is known to affect completions in other domains (Qi et al., 2024).

The challenge such an approach faces is that logprobs are non-deterministic in real-world LLM inference tasks.

## 2.2 NON-DETERMINISM IN LLMS

There are two main reasons for non-determinism in LLMs:

- **Intentional non-determinism**: The most common selection method for the next token in LLMs is temperature sampling. Given logits $z_i$ for each token $i$ in the vocabulary, the probability of sampling token $i$ at temperature $T$ is obtained from the softmax of the logits:

$$P(i, T) = \frac{e^{\frac{z_i}{T}}}{\sum_j e^{\frac{z_j}{T}}}. \tag{1}$$

  Logprobs are a unique representation of the logits, defined as the log of the probabilities at $T = 1$. In this paper, we are not concerned with non-determinism from temperature sampling, as we operate directly on the logprobs.

- **Unintentional non-determinism**: In practice however, logits often fluctuate between identical requests, resulting in a non-deterministic logprob vector. While the majority of inference kernels are run-to-run deterministic at the batch level, individual requests are non-deterministic, as they are affected by other requests in the batch (He & Lab, 2025). In addition, the software and hardware stack on top of which the model is deployed is also a source of non-determinism in production LLM APIs, where successive requests are likely to be routed to different GPUs (Zhang et al., 2024).

As an illustration of unintentional non-determinism, Figure 1 shows the logprobs returned over two weeks time by GPT-4.1, prompted with a single letter "x". The logprob of a given token isn't constant, but fluctuates around its average.

Given that logprobs are non-deterministic, we see each logprob returned as itself *sampled from a probability distribution*, and use standard hypothesis testing to assess whether the two distributions are the same.

## 2.3 TWO-SAMPLE TEST ON FIRST-TOKEN LOG-PROBABILITIES

Our approach, *logprob tracking* (**LT**), implements a hypothesis test to detect if the two distributions are the same, requiring minimal assumptions about logprob distributions. Namely, a permutation test based on the mean absolute distance between per-token average logprobs.

The overall procedure is shown in Figure 2: we send the same prompt to two LLM APIs, requesting a single token of output, repeating for $N$ samples. For each token in the returned logprob vectors, we

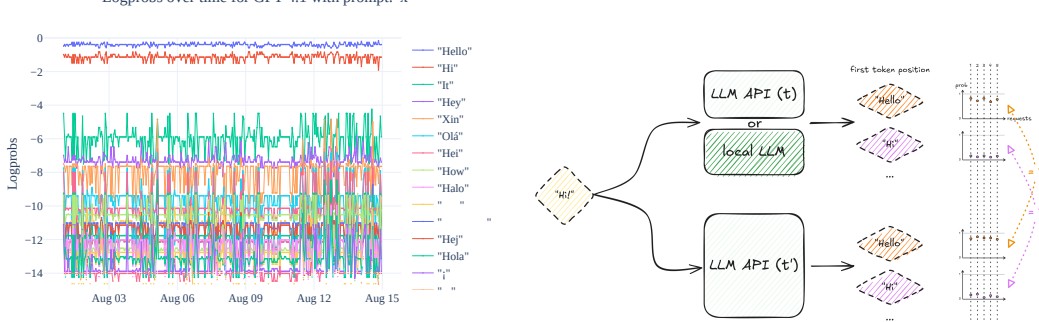

Figure 1: Logprob non-determinism in the wild: logprobs returned by the GPT-4.1 API, in August 2025.

Figure 2: Setup of our change detection method *logprob tracking* (**LT**).

obtain empirical measurements for its logprob value, and our goal is to compare these measurements to assess if the two distributions are the same.

Specifically, for an input prompt and two LLM APIs $API^{(1)}$ and $API^{(2)}$, we aim to compare their logprob distributions over the first response token, $P^{(1)}$ and $P^{(2)}$. We sample $N$ times from each distribution, where each sample returns only the top-$k$ tokens with their logprobs. Given this top-$k$ truncation, the list of tokens is not necessarily the same in each sample. When a token present in other samples is absent from a sample, we conservatively impute its missing logprob using the minimum logprob value from that sample, since we know that its logprob is not greater than this value.

Let $\mathcal{V} = \{t_1, \ldots, t_{n_{\text{tok}}}\}$ be the set of all tokens observed across both samples. Each of the $2N$ requests is represented as a vector of length $n_{\text{tok}}$, yielding matrices

$$T^{(1)}, T^{(2)} \in \mathbb{R}^{N \times n_{\text{tok}}}.$$

For each token $t_i$, we compute its average logprob:

$$\bar{a}_i^{(1)} = \tfrac{1}{N} \sum_{j=1}^{N} T_{j,i}^{(1)}, \qquad \bar{a}_i^{(2)} = \tfrac{1}{N} \sum_{j=1}^{N} T_{j,i}^{(2)}. \tag{2}$$

The test statistic is the average absolute distance between these means:

$$S = \frac{1}{n_{\text{tok}}} \sum_{i=1}^{n_{\text{tok}}} \left| \bar{a}_i^{(1)} - \bar{a}_i^{(2)} \right|. \tag{3}$$

A permutation test on the pooled samples is used to obtain a $p$-value for the null hypothesis that $P^{(1)}$ and $P^{(2)}$ are identical. For $b = 1, \ldots, B$ random permutations (each formed by splitting the pooled $2N$ samples into two groups of size $N$), compute permuted accuracies $\bar{a}^{(1),(b)}, \bar{a}^{(2),(b)}$ and the permutation statistic $S^{(b)}$ as previously.

The $p$-value is the probability of obtaining a larger test statistic than the observed one under the null hypothesis:

$$\hat{p} = \frac{1}{B} \sum_{b=1}^{B} \mathbf{1}\{S^{(b)} \geq S\}. \tag{4}$$

It can then be interpreted in relation with the chosen significance level $\alpha$. If $\hat{p} < \alpha$, we reject the null hypothesis and conclude that the two distributions are different. In our case, we conclude from $\hat{p} < \alpha$ that the two LLM APIs are not identical.

## 3 EXPERIMENTAL EVALUATION

We now evaluate LT and compare it to two competing baselines.

### 3.1 EVALUATION SETUP

**Creating variants of local models: TinyChange** A change detection method should ideally allow an auditor to configure a detection threshold based on the magnitude of the change that they want to detect. In particular, it should be able to distinguish between two different but close variants of a model. A benchmark creating variants of a model across a range of modification magnitudes should therefore be used to evaluate and compare detection methods, instead of evaluating on a few ad-hoc variants. As we are not aware of such a benchmark in the literature, we took the task to create one; we call it TinyChange. It is publicly available at https://github.com/timothee-chauvin/track-llm-apis.

This benchmark takes an LLM as input and generates a set of 58 variants, ranging from minor to more substantial modifications, reflecting practical operations a model might sustain in practice. Variants are produced across five levels of modification intensity, with difficulty increasing in powers of two. Specifically, we apply: **regular fine-tuning** and **LoRA fine-tuning** (Hu et al., 2022), each for 1 epoch with between 1 and 512 single-sample finetuning steps; **unstructured weight pruning**, either *by magnitude* or *by random selection*, removing a fraction of weights ranging from $2^{-10}$ up to 1; and **parameter noising**, adding independent Gaussian noise to each parameter with standard deviation $\sigma$ ranging from $2^{-15}$ to 1. More details in Appendix A.2.

The fine-tuning data is sampled from the LMSYS-Chat-1M dataset (Zheng et al., 2024), a dataset of real-world conversations between users and multiple LLMs, selecting only single-turn interactions with GPT-4. The goal is for the finetuning dataset to be both difficult to detect (because the data is in-distribution for models similar to GPT-4) and realistic (because most changes to production models are likely to be subtle).

We apply the TinyChange benchmark to 5 open-weight LLMs from 0.5B to 8B parameters: Qwen 2.5 0.5B, Gemma 3 1B, Phi-3 Mini 4k, Llama 3.1 8B, and OLMo 2 7B, resulting in a total of 290 variants.

**Evaluation Metrics** We consider the following three metrics:

1. **Cost**, measured as number of tokens required per hypothesis test.
2. **Discriminative power**, measured as the ROC AUC across different models and different variants per model, in the TinyChange benchmark.
3. **Sensitivity**, measured as the smallest modification that can be reliably distinguished from the original model, across several difficulty scales in TinyChange (number of steps of fine-tuning, fraction of weights removed in weight pruning, magnitude of the noise in random noise addition).

**Baselines** We compare logprob tracking (**LT**) to two state-of-the-art baselines:

- **MET** (*Model Equality Testing method*, Gao et al. (2025)) uses Maximum Mean Discrepancy (MMD) with a Hamming distance kernel to compare two distributions of responses. As in the original paper, we use $P_{MET} = 25$ prompts, 50-token responses, and $T = 1$.
- **MMLU-ALG** MMLU (*Massive Multitask Language Understanding*, Hendrycks et al. (2021)) is a well-known benchmark of the knowledge of LLMs. As the benchmark contains 14,042 questions, we restrict it to the "abstract_algebra" subset with $P_{MA} = 100$ questions, and call this baseline **MMLU-ALG**. Using MMLU is similar to the approach used in Chen et al. (2023), with the difference that MMLU consists of questions with multiple-choice answers A/B/C/D, thus considerably driving down the token cost of this approach compared to freeform answers. Each of the 100 prompts is sampled at $T = 0.1$, generating 5

---

A parameter-efficient fine-tuning method that introduces trainable low-rank matrices into pre-trained weights, enabling effective adaptation with far fewer trainable parameters

Huggingface IDs: Qwen/Qwen2.5-0.5B-Instruct, google/gemma-3-1b-it, microsoft/Phi-3-mini-4k-instruct, meta-llama/Llama-3.1-8B-Instruct, and allenai/OLMo-2-1124-7B-Instruct

output tokens. Similarly to 2.3, we define the two-sample test statistic as the mean absolute distance between per-question accuracies. More details in Appendix A.3.

**Sampling**    For each of the 5 original models and 290 variants ($V = 295$), we generate $S = 10,000$ samples per prompt using vLLM (Kwon et al., 2023). The MET and MMLU-ALG prompts are sampled at $T > 0$, where the non-determinism from temperature sampling far outweighs unintentional non-determinism. For LT however, we are concerned with the logprobs, so we introduce realistic noise from other requests by mixing our prompts with random traffic, also sampled from LMSYS-Chat-1M for simplicity. To generate each of the 10,000 samples, we mix and shuffle our $P_{LT} = 16$ tested prompts with 48 random prompts, and generate the logprobs for the first token of the batch with vLLM. (LT uses a single prompt, but we evaluated the influence of the prompt in 3.2.1). This results in $V \times S \times (P_{MET} + P_{MA} + P_{LT}) = 416$M outputs.

We then perform 1,000 *original/original* hypothesis tests on each original model, to obtain the distribution of the statistic under the null hypothesis, and 1,000 *original/variant* hypothesis tests on each variant. Each hypothesis test samples $N = 10$ outputs randomly from both distributions, for each prompt, and results in a test statistic.

For a given variant, we can finally compute the ROC curve and ROC AUC of a detection method from the *original/original* and *original/variant* statistics.

## 3.2    RESULTS

### 3.2.1    SHORT PROMPTS ARE SUFFICIENT FOR RELIABLE DETECTION

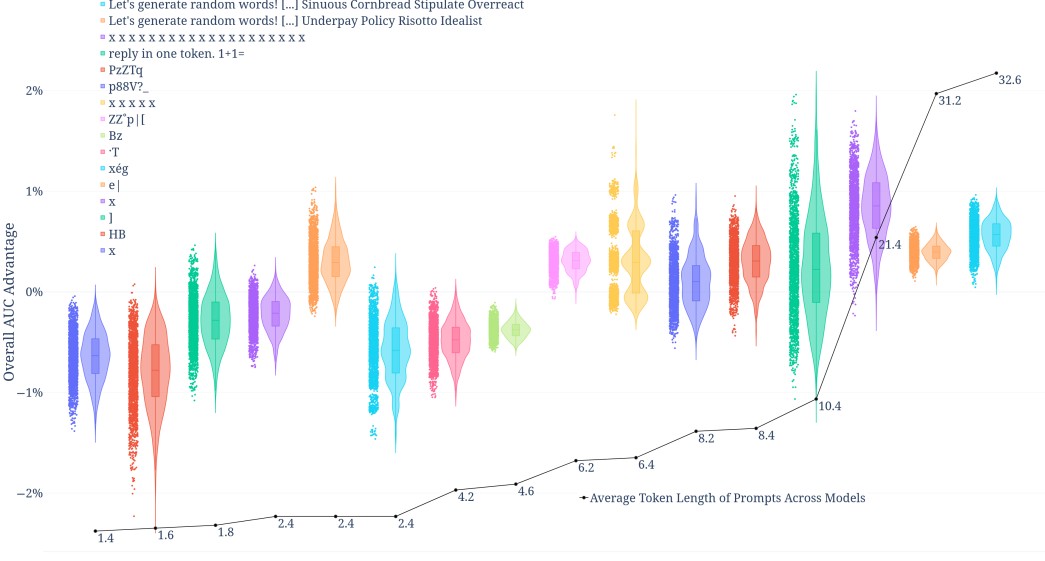

Figure 3: Minimal impact of the prompt length on the performance of LT. Average AUC across models and variants, 2,000 bootstraps at the model and test statistics levels.

Figure 3 shows each prompt's performance, relative to other prompts, across all models: for each model, we compute the absolute performance of all prompts on this model, as their ROC AUC. The relative performance of a prompt on a model is computed as the difference between its absolute performance and the average absolute performance of all prompts on this model. For each prompt, we finally compute the average of all its relative performances.

Prompts are ordered by their average token size across models. Longer prompts seem to perform slightly better, but by a small margin: about 1% difference in AUC between the shortest (1.5 tokens) and longest prompt (33 tokens), for a baseline of around 91% AUC (*cf* Table 1).

We conclude that very short prompts are sufficient to detect changes. In the rest of the experiments, we therefore use the shortest prompt, made of the single letter "x", which is 1 to 2 tokens long depending on the tokenizer.

### 3.2.2 LOGPROB TRACKING IS MORE SENSITIVE AND CONSIDERABLY CHEAPER THAN OTHER METHODS

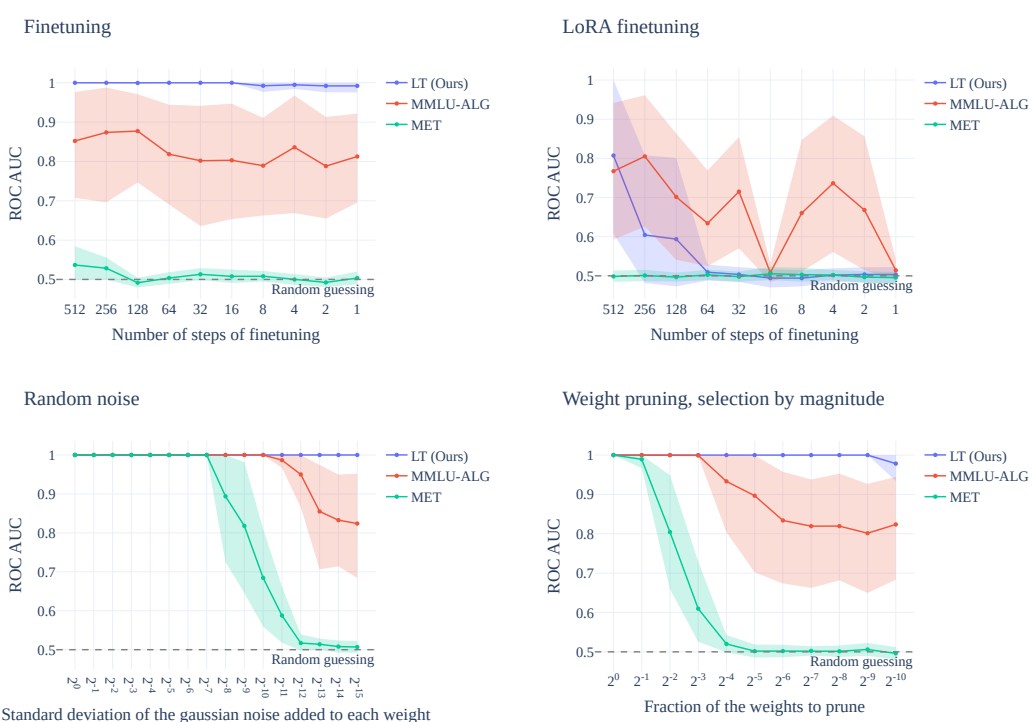

Figure 4: Average ROC AUC by difficulty level and method across LLMs (in each plot, difficulty increases from left to right). 95% CIs from 10,000 bootstraps at the model and test statistic levels.

Figure 4 plots the ROC-AUC of our approach (LT) and the two baselines (MMLU-ALG and MET) on four different types of changes included in the TinyChange benchmark (from left to right and top to bottom: Finetuning, LoRA Finetuning, Random noise, and Weight pruning), for different levels of change amplitude (x-axis). Logprob tracking clearly outperforms the baselines, with the possible exception of LoRA finetuning, which is challenging to detect for all methods.

If we consider a point estimate AUC of 0.9 as our threshold for good detection performance, we can compare the sensitivity of the different approaches, by considering the highest difficulties before which they reliably have good performance. In the weight pruning experiment, this same threshold difficulty is $2^{-1}$ for MET, $2^{-4}$ for MMLU-ALG, and $2^{-10}$ (or below) for LT. This represents a factor of $2^9 = 512$ between LT and MET, and slightly higher than $2^6 = 64$ between LT and MMLU-ALG.

We tentatively conclude that LT is 2-3 order of magnitude (OOMs) more sensitive to small changes than MET, and 1-2 OOMs more sensitive than our MMLU-ALG baseline.

### 3.2.3 CHANGE DETECTIONS IN REAL WORLD LLM APIS

**Methodology**  Due to the low cost of requesting single-token logprobs, we monitored 189 real-world LLM API endpoints from 10 providers hourly over more than 4 months, collecting more than 1.7M responses. We apply LT to this data to identify points of likely changes in APIs, using the test statistic $S$ (Eq. 3) as our detection signal.

| Model | MMLU-ALG | MET | LT (Ours) |
|---|---|---|---|
| Overall AUC (95% CI) | **0.878 (0.802, 0.944)** | 0.670 (0.612, 0.731) | **0.915 (0.864, 0.958)** |
| Token Count Per Test (I,O) | $(2.1 \times 10^5, 9.9 \times 10^3)$ | $(2.9 \times 10^4, 2.0 \times 10^4)$ | **(28, 20)** |
| Cost Per Year | $332 | $146 | **$0.14** |

Table 1: Overall ROC AUC across the TinyChange benchmark and across models, and token cost for each method. Cost per year of hourly sampling at GPT-4.1 pricing (input $3, output $12 / 1M tokens). 95% CIs from 10,000 bootstraps at the model, variant, and test statistic levels.

| Provider | Endpoints | Tracking Duration (Years) | Changes | Changes/Year |
|---|---|---|---|---|
| Fireworks | 14 | 4.1 | 9 | 2.2 |
| Lambda | 18 | 3.2 | 6 | 1.9 |
| Azure | 4 | 1.6 | 2 | 1.3 |
| Crusoe | 4 | 1.1 | 1 | 1.0 |
| Nebius | 36 | 8.2 | 7 | 0.9 |
| Chutes | 71 | 13.0 | 11 | 0.8 |
| xAI | 14 | 2.8 | 1 | 0.4 |
| OpenAI | 19 | 6.9 | 0 | 0.0 |
| Hyperbolic | 8 | 1.8 | 0 | 0.0 |
| Deepseek | 1 | 0.1 | 0 | 0.0 |
| **Total** | **189** | **42.8** | **37** | **0.86** |

Table 2: Change detections across providers. "Tracking Duration" refers to the cumulative duration of observation across endpoints.

Given the absence of ground truth, we flag a change only when the test statistic is high in absolute terms (to filter out small shifts) and relative to the endpoint's historical variance (to avoid false positives on high-variance endpoints).

Specifically, for a given endpoint, we compute its test statistic for each hourly query, by comparing the two adjacent 24-sample windows. We then compute the running mean and standard deviation of these test statistics, over a window of 100 samples. Finally, we report a change when the test statistic exceeds the running mean by at least 12 standard deviations, and is above 1.0.

**Results** We identified a total of 37 suspected changes, across 29 endpoints and 7 providers. Detailed statistics by provider are shown in Table 2.

Figure 5 shows the dates of detected changes across providers; detailed time series for each provider can be found in Appendix B.

We contacted some providers in June and July 2025, regarding earlier suspected changes, to ask if they could confirm that a change occurred, and describe the nature of the change. We did not hear back from xAI or Fireworks AI. Both Lambda and Nebius AI replied that they could not share details about suspected changes with us, with Nebius AI adding that they perform changes on a regular basis.

Notably, almost all detected changes (34 out of 37) affected open-weight models, where users might reasonably expect greater stability. This suggests that undisclosed changes are pervasive regardless of whether the model itself is open. This opacity in deployment undermines the transparency benefits of open weights.

## 4 LIMITATIONS AND FUTURE WORK

**Logprob support requirement** Our approach applies to APIs that support and return log probabilities. While this restricts its applicability compared to other methods leveraging only output tokens, we believe LT has value beyond its current applicability. First, our results demonstrate that logprob access significantly improves transparency, which we hope will motivate users to demand logprob support from providers, especially for open-weight models. Second, our work establishes a practical upper bound on detection sensitivity under inference non-determinism: we show that changes as small as a single fine-tuning step can be reliably detected by a black-box method. Third,

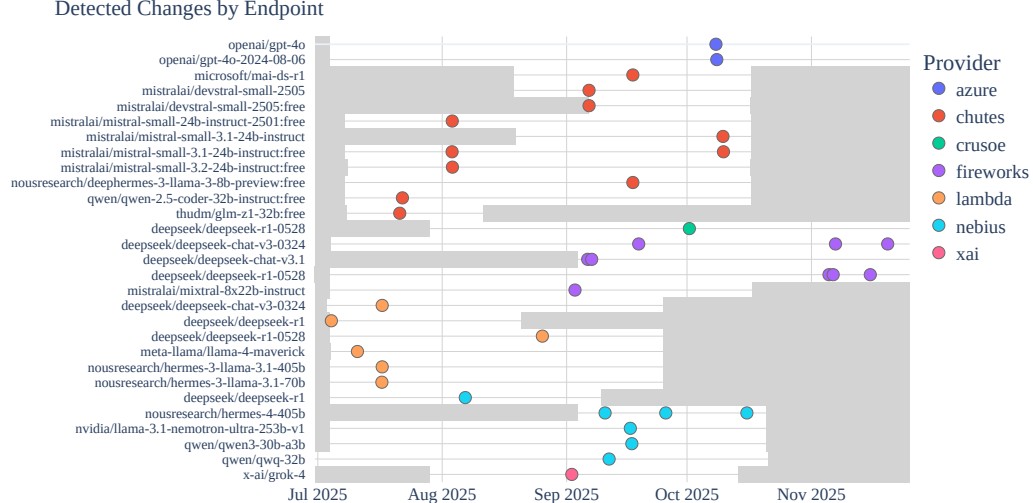

Figure 5: Dates of changes across providers and endpoints, queried hourly with the prompt "x". Periods where the endpoints weren't tracked are greyed out. 2 of the 37 changes (devstral-small-2505:free and mistral-small-3.1-24b-instruct:free) are also present in the non-free endpoints.

providers could integrate LT into their internal test suites for continuous monitoring of production inference.

**LT evasion**   Providers could be tempted to evade LT's detection by identifying monitoring queries and providing consistent responses only to these prompts. Alternatively, providers could cache logprobs to hide changes, though comprehensive caching is memory-prohibitive. Limited caching of short tokens and responses would force logprob tracking to rely on larger prompts, and therefore be more expensive. However, both of these techniques would risk creating detectable inconsistencies elsewhere and, if discovered, would cause greater reputational damage than undisclosed updates.

**LT obstruction**   A provider may also refuse to serve requests with only one or a few tokens of output. Indeed, we found that between 16:00 and 17:00 UTC on September 24, 2025, OpenAI started requiring at least 16 tokens of output on their GPT-4.1 line of models accessed via OpenRouter (though not on their own API).

**Single token of output**   Certain modifications may evade first-token detection. For example, providers could reduce verbosity during peak demand by adjusting generation-length parameters, such as the bias towards the end-of-sequence token, that leave initial tokens unchanged. However, these modifications are rare: the vast majority of realistic changes to an LLM API involve changing parameters that should be reflected in the first token of output.

**Limited information**   Our method does not distinguish between infrastructure changes and model updates, or give information on the exact nature of the change. While this may be viewed as a limitation, our opinion is that any systematic change affecting outputs can impact reproducibility, and should thus be disclosed. In addition, we view logprob tracking as a complement to existing audit methods—a low-cost, high-sensitivity approach that can run frequently and route alerts to more comprehensive investigations when changes are detected. While we focus on binary detection, our framework also naturally extends to quantifying change magnitude.

**Improvements**   We showed that using logprobs with a simple hypothesis testing method enables extremely sensitive change detection "out of the box", but detection performance can likely be increased further by tuning parameters. While the hypothesis testing framework simple, applicable,

---

at least for our one-token requests that requested output logprobs. The error message is: *"Invalid 'max_output_tokens': integer below minimum value. Expected a value $\geq 16$, but got 1 instead."*

and enables comparison with other methods, alternative approaches such as CUSUM (Page, 1954) control charts could also be explored.

## 5 RELATED WORK

**Change detection in ML APIs.** Prior to widespread LLM deployment, Chen et al. (2022) introduced methods to assess ML API shifts, and found significant shifts in a third of the ML APIs they tested.

**Change detection in LLM APIs.** Chen et al. (2023) compared versions of ChatGPT using multiple benchmarks, but this approach requires extensive testing across thousands of queries, and the versions they tested were explicitly not the same version. DailyBench (Phillips, 2025) was an actual attempt at benchmark-based continuous monitoring of LLM APIs, testing 5 LLMs 4 times a day on a version of HELMLite (Liang et al., 2023), but it was discontinued after 40 days due to costing $5 a day. Gao et al. (2025) compared output distributions with the Maximum Mean Discrepancy statistical test, using a Hamming distance kernel. Cai et al. (2025) re-implemented and compared multiple detection techniques.

**LLM fingerprinting.** Change detection and fingerprinting are closely related, as it's often possible to convert a change detection method into a fingerprinting method, and vice versa. However, some fingerprinting methods explicitly aim to be invariant to small changes in a model, or only attempt to identify an LLM within a closed set of known models. Pasquini et al. (2025) introduced a set of handwritten queries coupled with a classifier to identify which known model a given LLM is closest to. They explicitly aimed to produce queries that would be insensitive to small changes in a model, but sensitive to large changes. Sun et al. (2025) fine-tuned embedding models on LLM-generated texts, to identify which LLM out of a set of 5 generated a given text.

**Non-determinism in LLMs.** He & Lab (2025) demonstrated that run-to-run (i.e. on the same GPU) LLM non-determinism is due to a changing batch size based on the load on the inference server, proposing batch-invariant kernels as a solution. Zhang et al. (2024) leveraged within-platform determinism and cross-platform non-determinism to identify which hardware and software configuration a model is running on, also proposing a method based on logprobs.

**Formal verification of LLM outputs.** Formal verification is another approach for the problem we are concerned with: if an inference result can be proven to come from a specific model, then we can be confident that the model has not changed. zkLLM (Sun et al., 2024) introduced zero-knowledge proofs for LLMs. However, the proofs take several minutes to generate for a 13b LLM, and are on the order of 200kB in size, making this approach prohibitively expensive for high-performance inference servers. TOPLOC (Ong et al., 2025) used locality-sensitive hashing to considerably reduce the overhead of proving an inference result, but provider cooperation is still required.

## 6 CONCLUSION

Production LLM APIs are expected to be stable, providing their users—developers, researchers, regulators—the reliability and reproducibility that they require. We introduced logprob tracking (LT), an LLM API change detection method that treats first-token log probabilities as samples from a distribution and applies a two-sample permutation test to detect distributional shifts. LT reduces the cost of continuous monitoring by orders of magnitude compared to existing approaches while providing substantially higher sensitivity to small, realistic model modifications. We presented Tiny-Change, a benchmark of fine-grained model variants that quantifies sensitivity across several modification axes and difficulty scales, and used it to show that LT is a better detector of small changes—up to a single step of finetuning—than other methods, at a $1/1,000$th of the cost. We deployed LT in production environments at scale across hundreds of API endpoints and observed dozens of unexplained distributional shifts, illustrating that undocumented changes in deployed LLMs are both detectable and prevalent. LT provides a lightweight, practical first line of defense for reproducibility and integrity monitoring; it can be integrated into audit pipelines to trigger deeper, targeted investigations when a shift is detected.

## 7 REPRODUCIBILITY STATEMENT

The entirety of the source code we used in these experiments is available at https://github.com/timothee-chauvin/track-llm-apis. It can be used to reproduce the datasets, the TinyChange benchmark, the sampling outputs, and all plots. The exact versions of the dependencies used can be found in the file `uv.lock`. A `Dockerfile` is provided for the non-python dependencies. We ran the LLM variant generation and vLLM sampling on nodes of 2x H100 80GB GPUs.

## 8 ACKNOWLEDGMENTS

We acknowledge the support of the French Agence Nationale de la Recherche (ANR), under grant ANR-24-CE23-7787 (project PACMAM).

This work was supported by the Cluster SequoIA Chair FANG funded by ANR, reference number ANR-23-IACL-0009.

This project was provided with computing (AI) and storage resources by GENCI at IDRIS thanks to the grant 2025-AD011016369 on the supercomputer Jean Zay's H100 partition.

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

# A    IMPLEMENTATION DETAILS

## A.1    TESTING THE PREVALENCE OF LOGPROB SUPPORT ON OPENROUTER

The code to compute these statistics is in the file `logprob_prevalence_stats.py`.

We obtain the list of all 813 endpoints on OpenRouter by combining the APIs `v1/models` and `v1/model_id/endpoints`.

We first send a single prompt "x" to all of these endpoints, requesting a single token of output at temperature $T = 1$. We obtain a correct response from 710 endpoints (this number varies, as many of the errors are 429 rate limits, especially for the free models). We then send the same request to these reachable endpoints, this time requesting logprobs, and take note of which endpoints complied with this request.

We find 164 such endpoints (23%).

Fireworks AI and Azure return 5 logprobs, xAI 8 logprobs, and other providers return 20 logprobs.

In the case of OpenAI, we found that requesting less than 16 tokens of output from the GPT-4.1 series is not supported on OpenRouter, but only on the OpenAI API, as discussed in 4.

## A.2    TINYCHANGE HYPERPARAMETERS

In TinyChange, the source code of which is in the file `tinychange.py`, both finetuning and LoRA finetuning are performed with a learning rate $lr = 1 \times 10^{-6}$, with a single epoch. Each step of finetuning is performed on a single batch containing a single finetuning sample.

The LoRA hyperparameters are $r = 8$, $\alpha = 8$, dropout $= 0.0$, and the LoRA target modules are the attention layers' Key, Query, Value, and Output projections.

For most models, we patched their chat template to include the `{% generation %}` keyword and only finetune on assistant responses, instead of both questions and responses (see the file `chat_templates.toml`).

Model updates and inference are performed in BF16 precision.

## A.3    TWO-SAMPLE TEST ON THE MMLU BENCHMARK

We collect $N = 10,000$ independent responses from two LLMs, for each of $P = 100$ prompts, at temperature $T = 0.1$, using vLLM on an H100 80GB GPU.

We reduce the responses to a binary indicator of correctness: define the binary label for response $y_j, j \in \{1, \ldots, N\}$ of model $m \in \{1, 2\}$ on prompt $p \in \{1, \ldots, P\}$ as

$$I_{j,p}^{(m)} = \mathbf{1}\{y_{j,p}^{(m)}\text{is correct}\}. \tag{5}$$

For each prompt $p$ define the empirical accuracies

$$\bar{a}_p^{(1)} = \frac{1}{N}\sum_{j=1}^{N} I_{j,p}^{(1)}, \qquad \bar{a}_p^{(2)} = \frac{1}{N}\sum_{j=1}^{N} I_{j,p}^{(2)}. \tag{6}$$

The MMLU two-sample statistic is the mean absolute distance between per-prompt accuracies:

$$S_{\text{MMLU}} = \frac{1}{P}\sum_{p=1}^{P}\left|\bar{a}_p^{(1)} - \bar{a}_p^{(2)}\right|. \tag{7}$$

# B    LOGPROB TIME SERIES

Below, we share the logprob time series of a randomly selected endpoint for each provider. We prioritize showing an endpoint with detected changes; otherwise, a random endpoint is shown.

## B.1 WITH DETECTED CHANGES

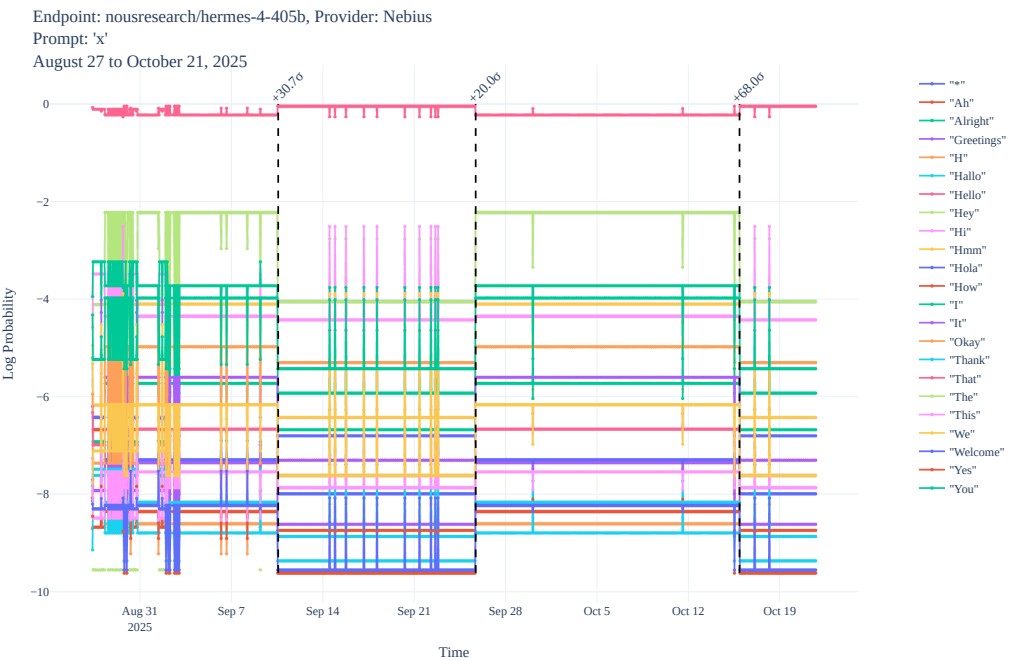

Figure 6: Nebius example: 3 changes detected on `hermes-4-405b` between August and October 2025.

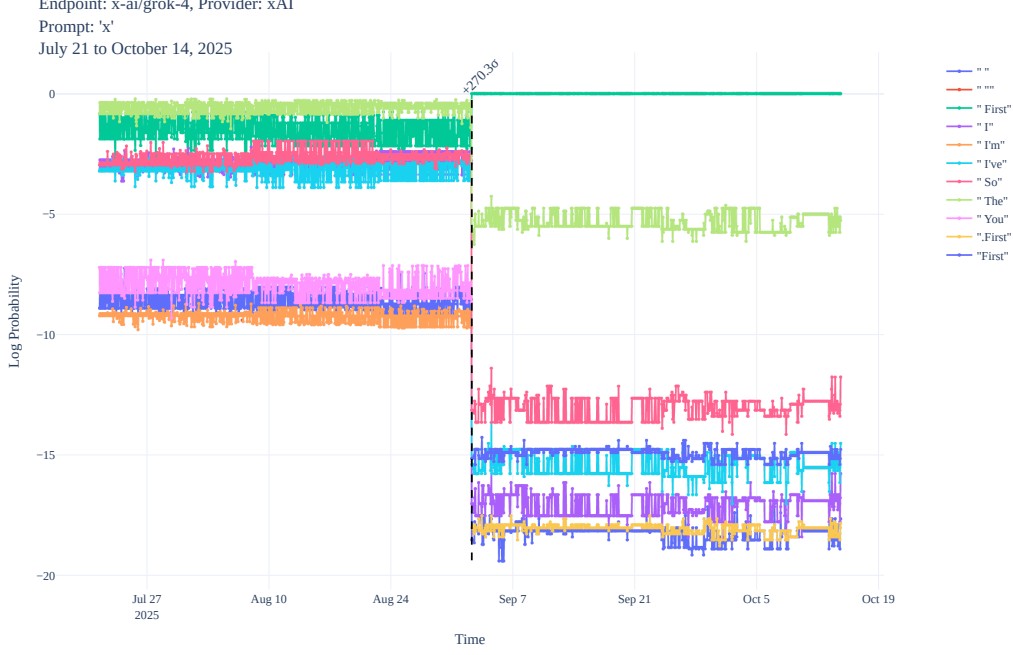

Figure 7: xAI example: 1 change detected on `grok-4` between July and October 2025.

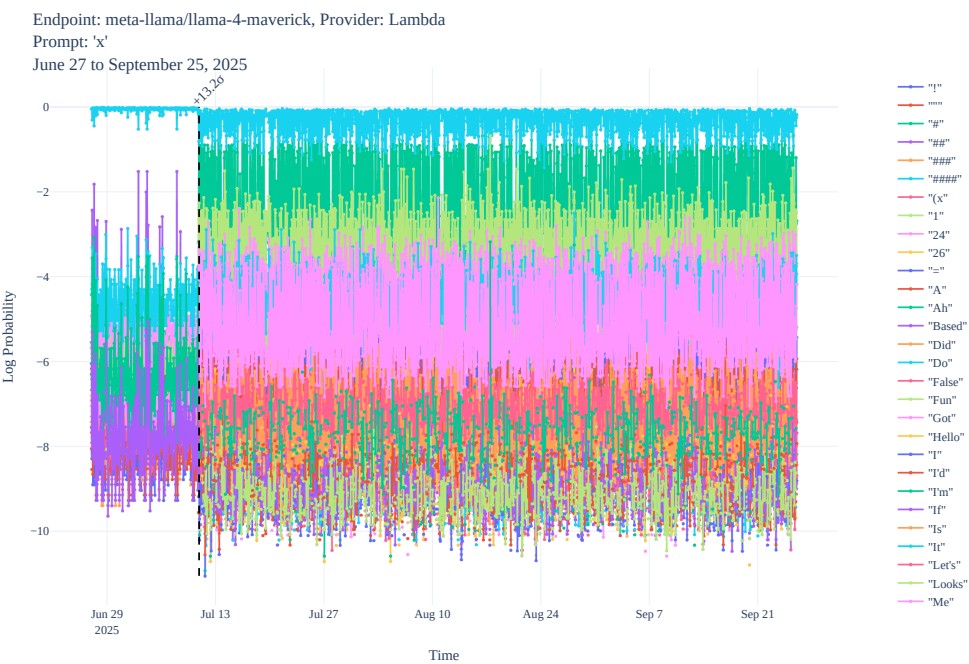

Figure 8: Lambda example: 1 change detected on `llama-4-maverick` between June and September 2025.

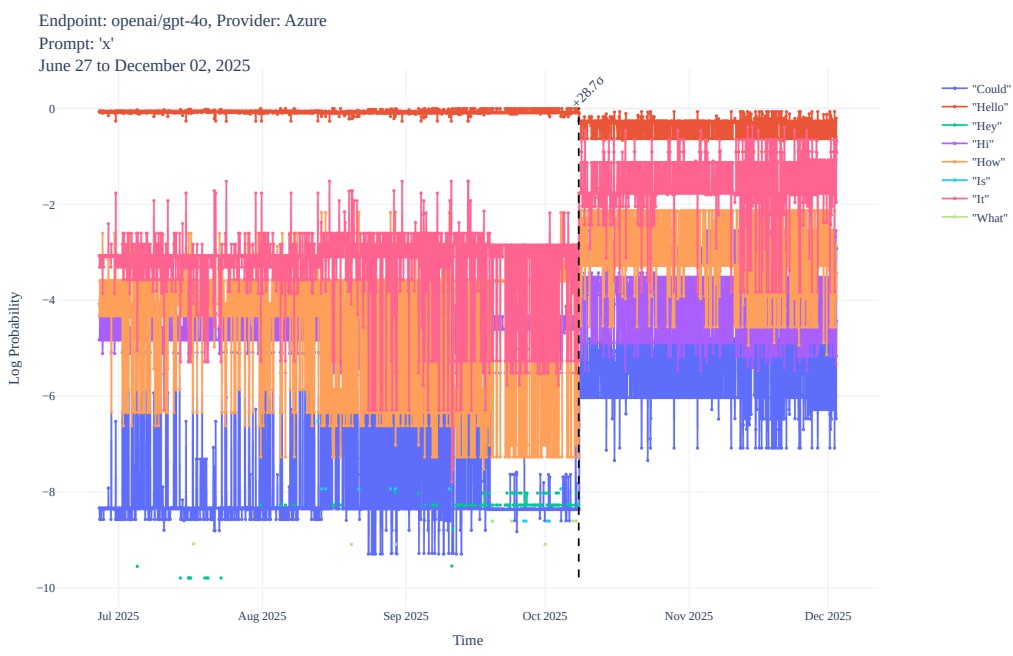

Figure 9: Azure example: 1 change detected on `gpt-4o` between June and December 2025.

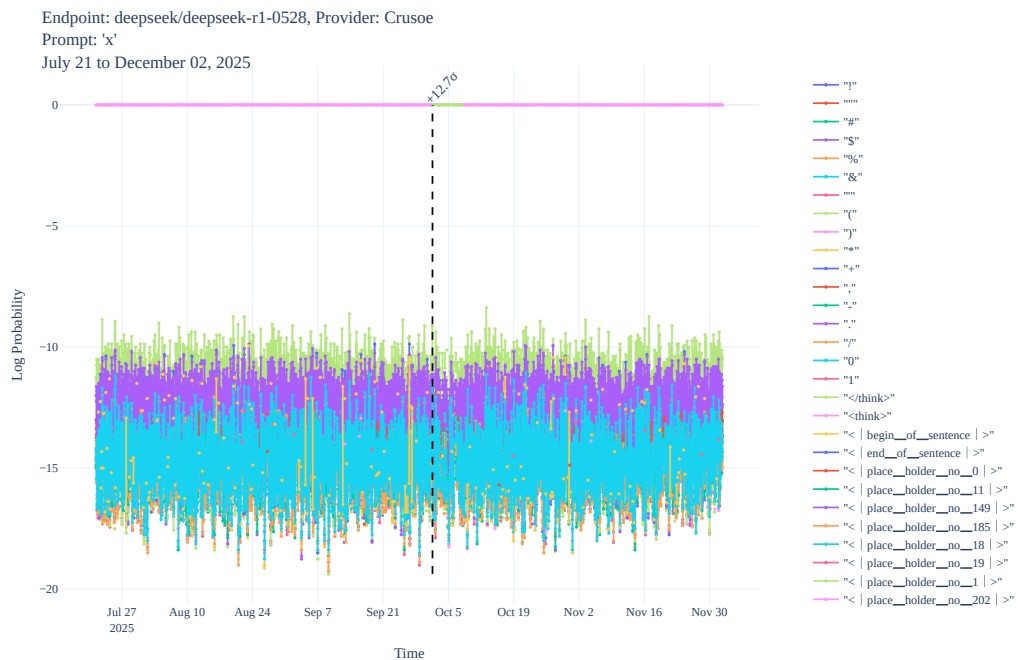

Figure 10: Crusoe example: 1 change detected on `deepseek-r1-0528` between July and December 2025 (see the substitution of the top token after the detected change point).

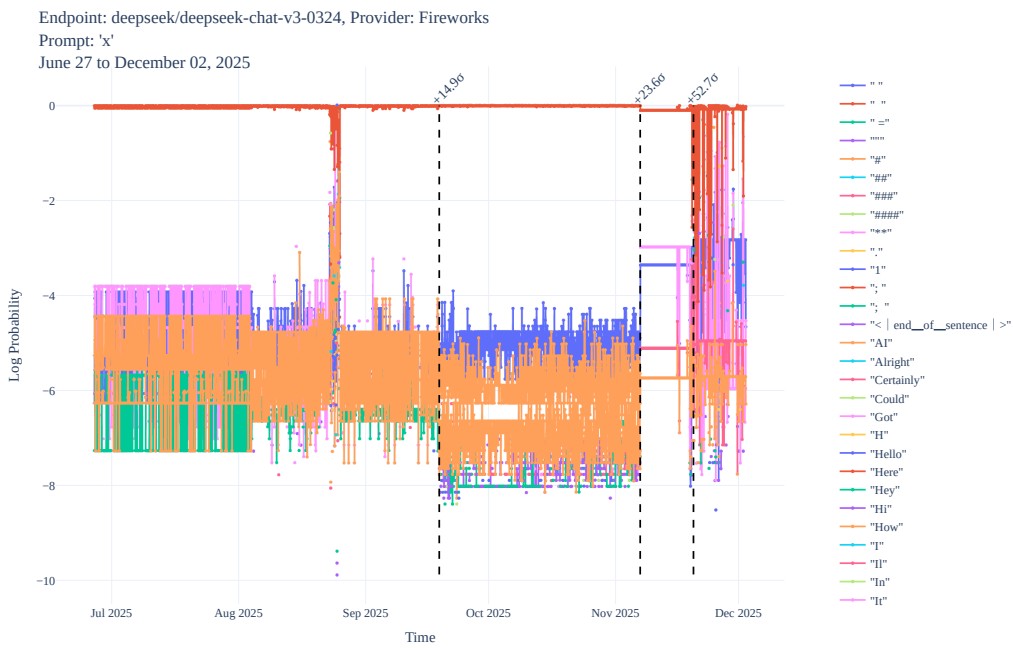

Figure 11: Fireworks example: 3 changes detected on `deepseek-chat-v3-0324` between June and December 2025.

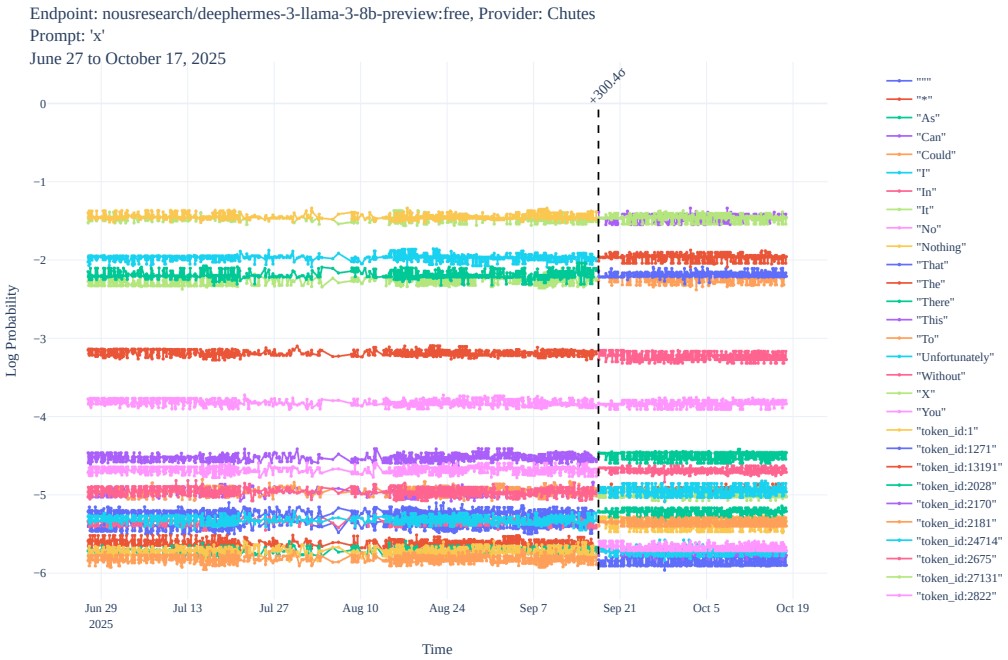

Figure 12: Chutes example: 1 change detected on `deephermes-3-llama-3-8b-preview:free` between June and October 2025. The decoded tokens changed while the probabilities themselves remained the same, probably reflecting a superficial change in the tokenizer.

## B.2 WITHOUT DETECTED CHANGES

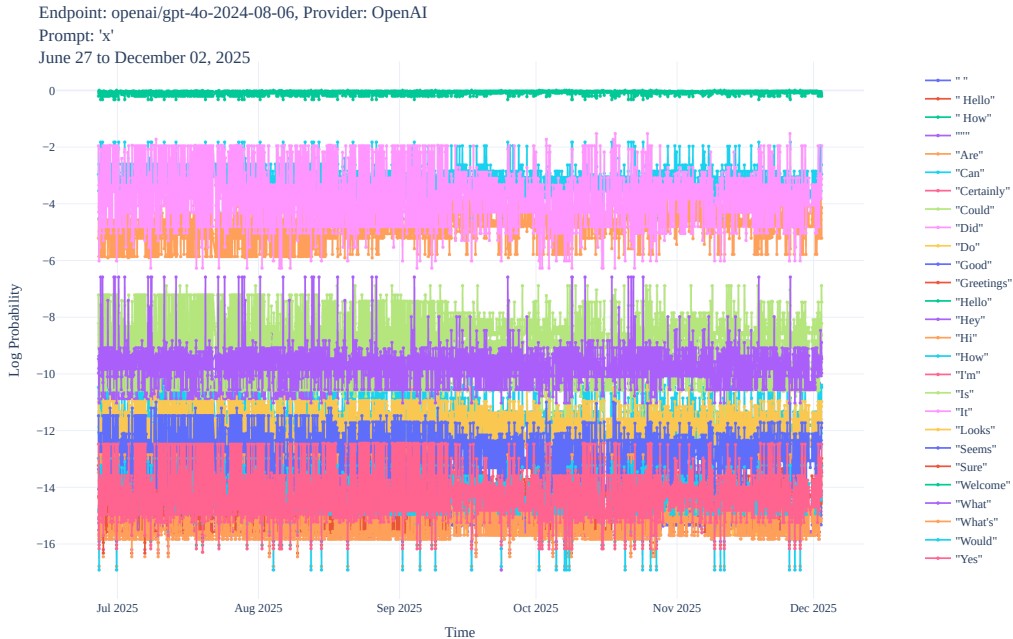

Figure 13: OpenAI example: no detected change on `gpt-4o-2024-08-06` between June and December 2025.

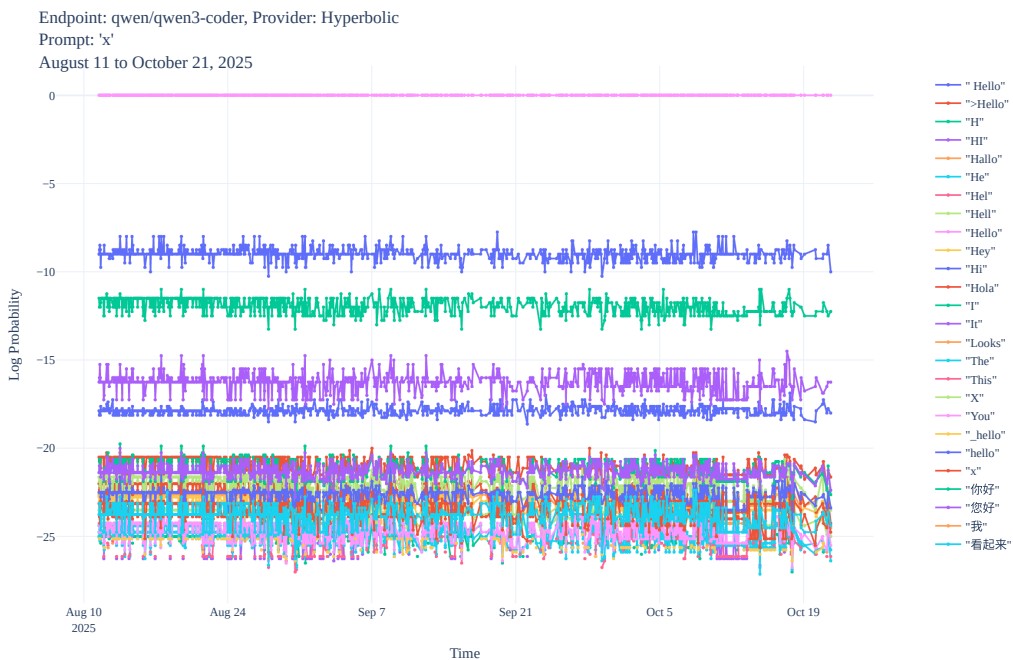

Figure 14: Hyperbolic example: no detected change on `qwen-3-coder` between August and October 2025.

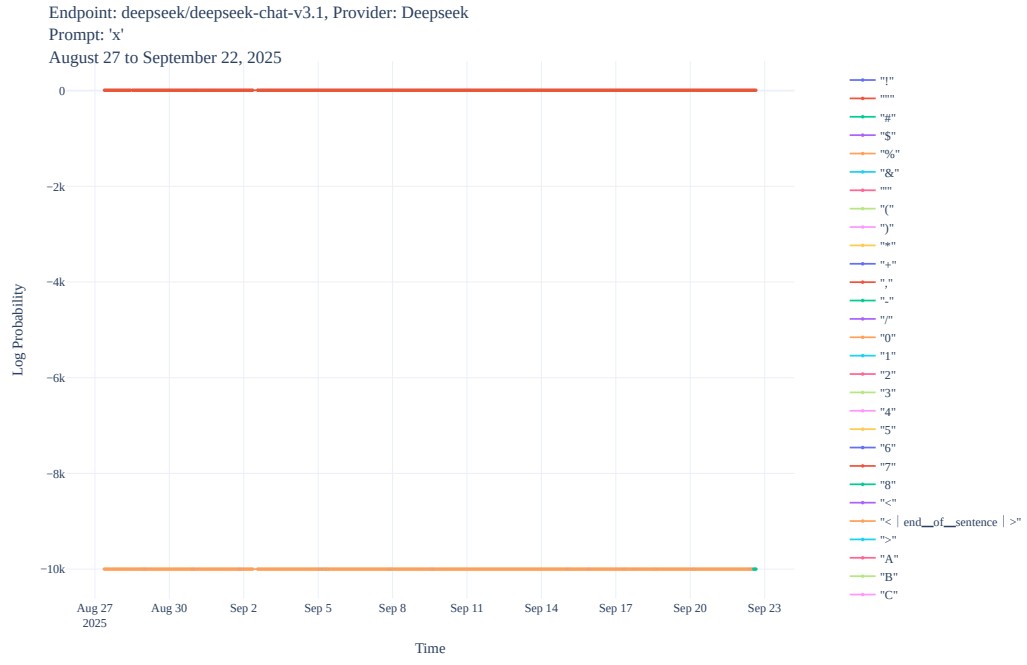

Figure 15: Deepseek example: no detected change on `deepseek-chat-v3.1` between August and September 2025.

## C  WHY IS LORA MORE CHALLENGING TO DETECT?

In figure 4, LoRA appears much more challenging to detect than regular finetuning, for the same number of steps of finetuning. Why is that?

An explanation is that LoRA produces much smaller model updates in terms of L2 distance between models, and we can think of this L2 distance as a proxy for the difficulty of detection.

|           | LoRA                  | Full                  |
|-----------|-----------------------|-----------------------|
| 1 step    | $4.3 \times 10^{-9}$  | $1.3 \times 10^{-5}$  |
| 16 steps  | $9.5 \times 10^{-8}$  | $1.3 \times 10^{-4}$  |
| 256 steps | $1.4 \times 10^{-5}$  | $3.5 \times 10^{-3}$  |

Table 3: L2 distances between original and finetuned Qwen2.5-0.5B-Instruct models.

Table 3 shows that LoRA updates are several OOMs smaller than regular updates, in terms of L2 distance.

We can make sense of these results by considering the geometry of the optimization: LoRA updates are constrained on a much smaller dimensional space than updates from full finetuning. If we consider the LoRA updates to be approximations of the full updates, the optimal approximation of the full update is its orthogonal projection on the LoRA subspace, which has a lower L2 norm.

Or more formally:

Let $A$ be the full weight update matrix obtained from regular fine-tuning, and let $L$ be the LoRA update matrix for the same MLP layer. Any real matrix $A$ can be decomposed into its Singular Value Decomposition as $A = U\Sigma V^T$, where $U$ and $V$ are orthogonal rotation matrices, and $\Sigma$ is a diagonal matrix containing the singular values $\sigma_i$ (positive and sorted descending), with non-null elements only on the diagonal. We have $\|A\| = \|\Sigma\|$.

The rank-$k$ approximation matrix $A_k$, constructed by retaining only the first $k$ diagonal elements of $\Sigma$ (the largest singular values) and setting all subsequent values to zero, is the best approximation of rank $k$ of $A$ in terms of minimizing the Frobenius (L2) distance to $A$. Its norm is:

$$\|A_k\| = \sqrt{\sum_{i=1}^{k} \sigma_i^2}. \tag{8}$$

Since $\sigma_i \geq 0$, this sum increases as $k$ increases. So the norm of the best approximation increases with rank $k$, and is smaller than the norm of the full update, which includes all singular values.

Now, we can see LoRA finetuning of rank $k$ as approximating this best approximation: $L \approx A_k$. Under the assumption of a good approximation, $L$ behaves similarly: $\|L\| < \|A\|$, with $\|L\|$ increasing with the LoRA rank.

As the rank of LoRA is typically much smaller than the rank of regular fine-tuning (in our case, $r = 8$ versus the number of parameters in each MLP), this explains why LoRA produces much smaller updates.

