# OpenReview forum: "Log Probability Tracking of LLM APIs"
_ICLR.cc/2026/Conference — ICLR 2026 Poster_

### Official Review · Reviewer_uCKu · 2025-10-25

**Soundness:** 3
**Presentation:** 3
**Contribution:** 1
**Rating:** 2
**Confidence:** 4

**Summary:**

This work addresses the problem of detecting changes in the model behind a blackbox LLM API. The authors propose detecting changes by monitoring the logprobs output by the API, which is possible for the ~23% of APIs that return logprobs. The key challenge is that logprobs can differ from call-to-call due to unintentional non-determinism. To address this, the authors propose logprob tracking (LT), which performs a permutation test on the difference between logrpobs. The authors find that this method achieves better detection AUC than baselines that do not use logprobs, while being less-costly to serve.

**Strengths:**

- The method is very simple to implement, when logprobs are available.
- The result that the detection AUC is not affected by prompt length is somewhat interesting.
- The authors release a benchmark for evaluating methods like this one.
- I like the plots showing how logprobs evolve over time.

**Weaknesses:**

- There is limited technical novelty in the methodology. Checking for differences in logprobs is the de facto approach for checking the correctness of language model implementations and APIs (*e.g.* from the VLLM tests https://github.com/vllm-project/vllm/blob/66a168a197ba214a5b70a74fa2e713c9eeb3251a/tests/models/utils.py#L90). It is well known that this is a more sensitive test than simply checking for text equality. That this is more effective than just checking text outputs is not a surprising result.
- I’m concerned about the significance of the problem and the practical utility of the method given the practices of frontier model providers today:
    - Most LLM APIs today (77%), do not return log-probs. Frontier model providers like OpenAI and Anthropic do not provide log-probs in responses. Given that the most popular API endpoints due not provide log-probs in response, it is unclear the practical utility of the method for most practitioners.
    - It’s unclear the degree to which the claim in the first sentence is true: “users of LLM APIs (developers, researchers, regulators) generally rely on the assumption that calling the same API endpoint will consistently serve the same model.” Most frontier LLMs are updated quite regularly (sometimes weekly or monthly), and most users are unaffected by the fact that the model is being updated. What should the user do if the model is updated? Frontier API providers do not provide the ability to use the old API.

**Questions:**

The method is not effective at detecting changes with LoRA fine-tuning. Why do you think this is?

---

> ### Author Response · Authors · 2025-11-15
>
> Thank you for your review and valuable feedback. Regarding the weaknesses and questions you brought up:
>
> ### W1 − Limited novelty
> **While the idea of comparing logprobs may be natural, working around implementation challenges is non-trivial:**
> - In the vLLM example you provided, the function `check_logprobs_close()` doesn't compare logprob values directly, but verifies that the *set of the token IDs of the top-k logprobs* matches (see e.g. [line 186](https://github.com/vllm-project/vllm/blob/66a168a197ba214a5b70a74fa2e713c9eeb3251a/tests/models/utils.py#L186)). This is presumably a way to work around non-determinism. But our approach of comparing the logprob values themselves is both significantly more precise, and tunable in terms of sensitivity threshold.
>
> - LT has not yet been explored in the literature. The closest work that we're aware of is [Cai et al., 2025](https://arxiv.org/abs/2504.04715v2) (section 4.1.6), but the authors conclude that logprob tracking is impractical due to two limitations:
>
>   - **non-determinism**: *"For auditors, this means logprob-based detection is vulnerable to false alarms unless requests are isolated or providers explicitly expose batch-invariant execution"* − we are the first to introduce a statistical test to work around this issue.
>   - **branch divergence in multi-token outputs**: *"Even at T = 0, greedy decoding can diverge after only tens of tokens, making logprob distributions very noisy"* − we are the first to suggest, and demonstrate, that requesting a single token of output is enough to detect even subtle changes, sidestepping this divergence issue.
>
> ### W2 − Reliance on logprobs
>
> Please refer to our discussion of this point in our top-level comment.
>
> ### W3 − Importance of stability
>
> Our paper focuses exclusively on LLM APIs, where users are typically more concerned about stability than on web interfaces − we will make this point clearer in the paper. These APIs are usually pinned to a particular model version (e.g. `gpt-4.1-2025-04-14`).
>
> We consider 3 notable categories of API users: **developers** want to avoid unexpected regressions in their applications; **researchers** seek reproducibility in their experiments; **regulators** perform safety testing on a particular model, and would like the API to remain consistent with the initial assessment.
>
> While many changes may come from good intentions, it's known that seemingly innocuous updates can impact LLM behavior in unrelated areas − see e.g. [Qi et al. (ICLR 2024)](https://openreview.net/forum?id=hTEGyKf0dZ).
>
> ### Q1 − Lower effectiveness on LoRA SFT
>
> We hypothesize this is due to LoRA SFT producing smaller model diffs than regular SFT, perhaps in terms of L2 distance between the models. We will investigate this in the next few weeks.

---

> ### Author Response · Authors · 2025-11-25
> **LoRA finetuning**
>
> Regarding your question on the lower performance on LoRA finetuning (Q1), we have found that **LoRA produces much smaller model updates than regular finetuning**, which explains its higher difficulty.
>
> On Qwen/Qwen2.5-0.5B-Instruct, the L2 distances between the original and finetuned models are the following, depending on the number of steps of finetuning:
>
> |           | LoRA   | full   |
> |-----------|--------|--------|
> | 1 step    | 4.3e-9 | 1.3e-5 |
> | 16 steps  | 9.5e-8 | 1.3e-4 |
> | 256 steps | 1.4e-5 | 3.5e-3 |
>
> We see that LoRA updates are several OOMs smaller than regular updates, in terms of L2 distance. And we can think of the L2 distance between models as a proxy for the difficulty of detection.
>
> We can make sense of these results by considering the geometry of the optimization: LoRA updates are constrained on a much smaller dimensional space than updates from full finetuning. If we consider the LoRA updates to be approximations of the full updates, the optimal approximation of the full update is its orthogonal projection on the LoRA subspace, which has a lower L2 norm.
>
> Or more formally:
>
> Let $A$ be the full weight update matrix obtained from regular fine-tuning, and let $L$ be the LoRA update matrix for the same MLP layer. Any real matrix $A$ can be decomposed into its Singular Value Decomposition as $A = U \Sigma V^T$, where $U$ and $V$ are orthogonal rotation matrices, and $\Sigma$ is a diagonal matrix containing the singular values $\sigma_i$ (positive and sorted descending), with non-null elements only on the diagonal. We have $\| A \| = \| \Sigma \|$.
>
> The rank-$k$ approximation matrix $A_k$, constructed by retaining only the first $k$ diagonal elements of $\Sigma$ (the largest singular values) and setting all subsequent values to zero, is the best approximation of rank $k$ of $A$ in terms of minimizing the Frobenius (L2) distance to $A$. Its norm is:
>
> $\| A_k \| = \sqrt{\sum_{i=1}^k \sigma_i^2}$.
>
> Since $\sigma_i \geq 0$, this sum increases as $k$ increases. So the norm of the best approximation increases with rank $k$, and is smaller than the norm of the full update, which includes all singular values.
>
> Now, we can see LoRA finetuning of rank $k$ as approximating this best approximation: $L \approx A_k$. Under the assumption of a good approximation, $L$ behaves similarly: $\| L \| < \| A \|$, with $\| L \|$ increasing with the LoRA rank.
>
> As the rank of LoRA is typically much smaller than the rank of regular fine-tuning (in our case, $r=8$ versus the number of parameters in each MLP), this explains why LoRA is much more challenging to detect than regular finetuning − making it a useful addition to the benchmark. We will add the above explanation to the paper.
>
> Regarding the apparent peaks in AUC of our MMLU-ALG baseline on 2-8 steps of LoRA, we note that the non-monotonic pattern (with chance-level performance at 16 steps) is inconsistent with true detection capability. We hypothesize that these anomalous peaks are statistical noise due to the stochasticity inherent in fine-tuning.

---

### Official Review · Reviewer_k7Vu · 2025-10-31

**Soundness:** 3
**Presentation:** 3
**Contribution:** 2
**Rating:** 6
**Confidence:** 4

**Summary:**

This paper prsents Logprob Tracking (LT), a technique for LLM drift detection under resource constraints. The key idea is to sample the log probability of the first generated token, and then perform a permutation test. This allows LT to detect minor model updates with 1000x lower cost than existing methods. The authors also propose a benchmark to test small model modification and conduct experiments on controllable model drifts and real-world API drifts.

**Strengths:**

- Addresses an important and underexplored reproducibility problem: LLM APIs are increasingly widely applied while continously updated, and monitoring these behavior shift is an important topic.

- Simple, cost-efficient, and elegant approach: The proposed technique is easy to understand and implement: it basically tests if the distribution of the log probability of the first generated token has changed or not, using a permutation test. This is neat, and also cost-efficient, as it only requires the first token of a given query.


- New benchmark (TinyChange) for fine-grained change detection: This benchmark offers a systemtical way to generate model drift with different magnitudes.

**Weaknesses:**

- Depends on APIs exposing logprobs: As the authors notice too, only a small fraction of existing API providers (~23% in openrouter) offers logprob access.

- Detections w/o directions: The proposed method only detects whether a change occurred, not what changed or how the change looked like. In particular, it is unclear if the change leads to better responses to user queries, or what kind of biases or skills were introduced or forgotten in the model update. In practice, this is often more important than just detecting a change.


- Limited evaluation and analysis of real-world APIs: Section 3.2.3 briefly mentions drift detection of real-world APIs and the model providers' responses. A more detailed analysis on real-world API drift would improve the paper a lot and make it more relevant to practical senarios.

**Questions:**

- How robust is LT when the logprob sampling temperature or top-k cutoff changes across time?

- Could LT be extended to detect which type of change (e.g., quantization vs. fine-tuning) occurred?

- Have the authors considered multi-token extensions or dynamic prompting to improve evasion resistance?

- Can TinyChange be used to evaluate fingerprinting methods as well, given the conceptual overlap?

---

> ### Author Response · Authors · 2025-11-15
>
> Thank you for your review and valuable feedback. Regarding the weaknesses and questions you brought up:
>
> ### W1 − Reliance on logprobs
>
> Please refer to our discussion of this point in our top-level comment.
>
> ### W2 − Detections without directions
>
> To the best of our knowledge, no single notion of "direction" adequately captures the diverse requirements for model stability across contexts: developers need non-regression on application-specific benchmarks, regulators track safety metrics, and researchers have experiment-specific needs. Note also that the MET baseline has the same limitation.
>
> ### W3 − Limited evaluation on real-world APIs
>
> This is an excellent point, and we plan to get back to you with more analysis of our collected responses from real-world APIs as soon as possible.
>
> ### Q1 − LT robustness
>
> As the logarithm of the probabilities, logprobs could in theory depend on the temperature, but in practice the term is used to refer to `log(probabilities(T=1))`, and API providers follow this convention. We will clarify this point in the paper.
>
> Regarding the top-k cutoff: we haven't seen any case of the top-k cutoff changing over time for a given API endpoint. However, while 20 is the most common number of logprobs returned, a few providers return a lower number of logprobs (e.g. the Grok API returns 8). If you think this would improve the paper, we could complement the analysis with an impact assessment of the number of available logprobs.
>
> ### Q2 − More info on changes
>
> Please refer to our response in our top-level comment.
>
> ### Q3 − Multi-token extensions
>
> We considered using multiple output tokens, but a single token already provided very high sensitivity, so we did not pursue it further. Branch divergence (see our [response to reviewer uCKu, section W1](https://openreview.net/forum?id=hFxivbAgVP&noteId=Xg1jxFR0Pb)) would make multi-token analysis more challenging to implement. It's also not clear that allocating tokens to longer responses would be more cost-effective than the simpler approach of sending multiple 1-token requests with different prompts, especially since output tokens are often more expensive than input tokens.
>
>
> ### Q4 − Dynamic prompting
> We haven't looked into dynamic prompting beyond our "LT evasion" paragraph in Section 4, but this could be investigated by future work.
>
> ### Q5: TinyChange for fingerprinting methods
>
> This is a great suggestion. TinyChange could indeed be used wherever small, controlled differences between models are needed. This would apply in particular to fingerprinting methods, especially those that can work on models not seen ahead of time (i.e. open-set rather than closed-set).
>
> A possible limitation would be if the fingerprinting method explicitly attempts to be insensitive to small changes, so that small variants of a model share the same signature, like LLMMap ([Pasquini et al, 2025](https://arxiv.org/abs/2407.15847)).
>
> Overall, we would be excited if TinyChange could support research into fingerprinting methods that are more sensitive than those currently available.

---

### Official Review · Reviewer_pLmm · 2025-11-04

**Soundness:** 3
**Presentation:** 4
**Contribution:** 4
**Rating:** 8
**Confidence:** 3

**Summary:**

The paper introduces Logprob Tracking (LT), a cost-effective and highly sensitive method for continuously monitoring LLM APIs for unintended or undisclosed changes.  Users rely on LLM APIs remaining consistent, yet providers frequently implement model modifications that go unmonitored. LT addresses this by exploiting the log probabilities of the first output token, which provide a significantly richer information source than the generated tokens alone. To overcome the non-deterministic nature of log probabilities in production environments, LT employs a simple two-sample permutation test based on the average absolute distance between per-token mean log probabilities. Experiments on the TinyChange benchmark show LT can detect changes as small as a single fine-tuning step, while achieving cost reductions of up to 1,000 times compared to baselines like MET and MMLU-ALG.

**Strengths:**

LT drastically reduces the cost of continuous monitoring, achieving sensitivity gains at a cost that is up to three orders of magnitude cheaper than competing state-of-the-art methods

LT provides substantially higher discriminative power and sensitivity than existing approaches. It reliably detects small modifications such as a single step of fine-tuning and demonstrates detection performance for weight pruning at an amplitude 512 times smaller than MET.

The authors use permutation tests on the mean absolute difference of per-token average log probabilities to handle the inherent non-determinism observed in production APIs.  This is a nice addition that addresses a key limitation in monitoring production systems.  Because the permutation test only requires minimally acquired data from a single output token from a 1-token input prompt, the overall cost of monitoring is drastically reduced.

**Weaknesses:**

The entire methodology is contingent on the API provider supporting and returning log probabilities. Data presented in the paper indicates that only 23% of reachable endpoints on OpenRouter support this.  This limits the applicability of the approach.

LLM providers can obstruct LT by requiring minimum output token lengths

 The reliance on log probabilities for only the first output token might miss certain modifications such as adjusting the generation-length parameter.

**Questions:**

What mechanisms, beyond binary detection, can be integrated into the LT framework to help auditors diagnose the likely source or severity of the detected change (e.g., distinguishing an infrastructure update from a behavioral fine-tuning step)?

How does LT perform against variants created in the TinyChange benchmark where only the EOS token log probability bias is subtly modified, rather than the core model weights?

---

> ### Author Response · Authors · 2025-11-15
>
> Thank you for your review and valuable feedback. Regarding the weaknesses and questions you brought up:
>
> ### W1 − Reliance on logprobs
>
> Please refer to our discussion of this point in our top-level comment.
>
> ### W2 − Minimum output token lengths
>
> In this paper we focus on non-adversarial LT, leaving adversarial platform behavior to future work. In this particular case, while enforcing a minimum number of output tokens could indeed make LT more expensive, we believe this would defeat the point of providing logprobs, as developers often use them for single-token answers, e.g. in MCQ benchmarks.
>
> ### Q1 − More info on changes
>
> Please refer to our response in our top-level comment.
>
> ### Q2 − EOS token logit bias detection
>
> This is an interesting question: pricing based on monthly subscriptions incentivizes providers to favor shorter responses (increase the EOS logit), while per-token pricing − used by all LLM APIs we're aware of − incentivizes them to favor longer responses (decrease the EOS logit).
>
> If this EOS logit bias is present regardless of the token position, this can in principle be detected by LT. So if time permits, we will try to add this experiment in the next few weeks.

---

### Author Response · Authors · 2025-11-15

We would like to thank all three reviewers for their constructive feedback. Based on the comments, we will revise the paper in the coming weeks by clarifying several points and adding additional experiments.

Below, we respond to concerns and questions shared between several reviewers.

### Reliance on API providers supporting logprobs

As pLmm, k7Vu and uCKu noted, LT is contingent on API providers supporting and returning logprobs, which currently represents a minority of providers.

This is an important point, and we haven't discussed it sufficiently in the paper.

We agree with this limitation, but we believe that LT has value regardless of the current prevalence of logprob support:

- **motivating greater transparency**: API providers always have the option of returning logprobs, and our results show that doing so significantly improves transparency in the LLM ecosystem. We also found that many providers perform undisclosed changes that alter model outputs, including on open-weight models. We hope this will motivate users to demand logprob support and change disclosure from API providers, and to be suspicious of those that refuse. This is especially important for open-weight models, which have been praised for increasing transparency, yet are often accessed through APIs that remain unaudited.

- **upper bound on other methods' effectiveness**: We show for the first time that under inference non-determinism, changes as small as a single step of fine-tuning can be reliably detected by a black-box method. This makes our approach a practical “upper bound” baseline for techniques that do not rely on logprobs, and our TinyChange benchmark enables researchers to rigorously evaluate the sensitivity of their own methods.

- **possible internal use**: even if providers don't expose logprobs publicly, they might use this method as part of their internal continuous monitoring methods.

We will add these points in the paper.

### Extending LT beyond binary detection

> **k7Vu**: Could LT be extended to detect which type of change (e.g., quantization vs. fine-tuning) occurred?

> **pLmm**: What mechanisms, beyond binary detection, can be integrated into the LT framework to help auditors diagnose the likely source or severity of the detected change (e.g., distinguishing an infrastructure update from a behavioral fine-tuning step)?

Regarding the *severity* or *magnitude* of the change, the LT test statistic can be a good indicator − the larger the change, the larger the statistic should be. If time permits, we will validate this relationship experimentally.

Detecting the *source* of the change is more challenging. While different modification types might leave distinctive signatures in the resulting logprobs, this is not guaranteed. In fact, we are not aware of any black-box method that can identify the cause of the difference between two LLMs without knowing the LLM and a list of possible changes in advance.

So our approach is to focus on the *impact* of the change rather than its source: with LT, we aim to provide the first method that can detect changes at scale, allowing to alert API users when a model has changed. Once notified, users − who best understand their own applications − can evaluate how the change affects their particular use case.

---

### Author Response · Authors · 2025-11-26

Following up on our previous responses, we have uploaded a new revision with changes highlighted in blue. Key updates:
* Improved the presentation and argumentation in several parts of the paper, by better highlighting the importance of API stability (reviewer uCKu), and the value that LT provides beyond its current applicability (reviewers uCKu, pLmm, k7Vu).
* Added a large-scale analysis of real-world API data, as suggested by reviewer k7Vu. We identified 37 undisclosed changes across 7 of the 10 providers monitored. Notably, 34 of these 37 changes affected open-weight models. Full time series for each provider, with detected change points marked, have been added to the appendix. To our knowledge, this is the first large-scale empirical demonstration of undisclosed changes in LLM APIs.

---

### Author Response · Authors · 2025-12-02
**Summary for Area Chair**

Dear AC,

Below is a summary of the reviews and our responses. Our changes in the new revision are highlighted in blue.

Reviewers liked our method's simplicity, discriminative power, low cost, and handling of the non-determinism in real-world LLM APIs. They also appreciated the release of our benchmark, TinyChange, to systematically generate model drift with different magnitudes.

We responded to the weaknesses they brought up as follows:

- **Reliance on logprobs (all reviewers)**: We clarified why LT has value regardless of current logprob support rates: motivating greater transparency from providers, establishing an upper bound on other methods' effectiveness, and enabling internal monitoring by providers.

- **Limited evaluation of real-world APIs (reviewer k7Vu)**: reviewer k7Vu wrote: *"A more detailed analysis on real-world API drift would improve the paper a lot and make it more relevant to practical scenarios"*. We added such an analysis based on our collected data, and identified 37 undisclosed changes across 7 of 10 monitored providers. Notably, 34 of these 37 changes affected open-weight models. Full time series with detected change points are in the appendix. To our knowledge, this is the first large-scale empirical demonstration of undisclosed changes in LLM APIs.

- **Limited novelty (reviewer uCKu)**: reviewer uCKu wrote: *"Checking for differences in logprobs is the de facto approach for checking the correctness of language model implementations and APIs (e.g. from the VLLM tests https://github.com/vllm-project/vllm/blob/66a168a197ba214a5b70a74fa2e713c9eeb3251a/tests/models/utils.py#L90)"*. Disagreeing with this claim, we noted that in the provided vLLM example, the code only compares *the sets of the top-k token IDs* (per the function docstring: *"set of highest-logprob token ids must match between seq0 and seq1 at all sampled token offsets"*). This is significantly less precise than our method, and isn't tunable in terms of sensitivity threshold. We also mentioned that the closest prior attempt at leveraging logprobs ([Cai et al., 2025](https://arxiv.org/abs/2504.04715v2)) concluded that doing so was impractical, due to non-determinism and branch divergence − both of which we address with our permutation test and single-token approach.

- **Importance of stability (reviewer uCKu)**: We clarified that our paper focuses on LLM APIs, which are typically pinned on specific versions (e.g. `gpt-4.1-2025-04-14`), and where users are more concerned about stability than on web interfaces: **developers** want to avoid unexpected regressions in their applications; **researchers** seek reproducibility in their experiments; **regulators** want consistency with initial safety assessments. We also noted that while many changes may come from good intentions, seemingly innocuous updates can impact LLM behavior in unrelated areas ([Qi et al., 2024](https://openreview.net/forum?id=hTEGyKf0dZ)).

We also responded to reviewer uCKu's question on LoRA detection difficulty with an analysis showing that LoRA updates have several OOMs smaller L2 distance from the original model than regular fine-tuning.

---

### Meta-Review · Area_Chair_g25f · 2026-01-06

**Summary:**

The submission "Log Probability Tracking of LLM APIs" argues that logprobs, when available, are the most straightforward and optimal way to detect quality fluctuations of models hosted by various API providers. In the revised version of the draft, the authors show that they use this setup to monitor 189 provider endpoints on an hourly interval, and are able to detect 37 changes in serving quality.

**Reviewer Concerns:**

There are a number of fundamental limitations with the approach proposed in this submission brought up by the reviewers (which are partly discussed in the limitations section).
1. Only APIs that provide logprobs can be tracked. This is ultimately the largest limitation. Most providers do not provide logprobs as these can be used to exfiltrate model details (e.g. Carlini et al. "Stealing Part of a Production Language Model", which is mentioned in passing in the submission).
2. Only changes to prefill can be tracked. As the proposed approach, to my understanding, only queries a single token, it can only be used to evaluate changes in prefill strategies. Changes in generation, e.g. using KV-cache compression/quantization, or speculative decoding, or problems with KV states (could problems like https://www.anthropic.com/engineering/a-postmortem-of-three-recent-issues even be tracked on a similar API that provided logprobs?).
3. Adversarial behavior by providers. Given that only the first token is tracked, providers may be incentivized to provide it at a higher, or more stable quality, if this kind of testing becomes more common.

Nevertheless, I do think the simplicity of the proposed tracking strategy is a strength, and, while the proposed approach is, of course, obvious in hindsight, cheap&continual tracking of public-facing logprob APIs is a good idea where logprobs are available. Due to this, I recommend poster acceptance.

**Reviewer Scores:**

pLmm no changes, k7Vu no changes likely, uCKu may increase to 4, but probably not likely.

---

### Decision · Program_Chairs · 2026-01-26

Accept (Poster)